# Hematological and Biochemical Effects Associated with Prolonged Administration of the NSAID Firocoxib in Adult Healthy Horses

**DOI:** 10.3390/vetsci11060256

**Published:** 2024-06-05

**Authors:** Fernanda Saules Ignácio, Luana Venâncio Garcia, Giovanna Gati de Souza, Lidiana Zanetti Amatti, Luiz Daniel de Barros, Don R. Bergfelt, Giovana Siqueira Camargo, Cezinande de Meira, Breno Fernando Martins de Almeida

**Affiliations:** 1Department of Veterinary Surgery and Animal Surgery, School of Veterinary Medicine and Animal Science (FMVZ), Sao Paulo State University (UNESP), Botucatu 18618-681, SP, Brazil; giovana.camargo@unesp.br (G.S.C.); c.meira@unesp.br (C.d.M.); 2Department of Veterinary Medicine, University Center of the Integrated Faculties of Ourinhos (Unifio), Ourinhos 19909-100, SP, Brazil; luanagarcia_venancio@hotmail.com (L.V.G.); giovannagati@hotmail.com (G.G.d.S.); lidianaamatti@hotmail.com (L.Z.A.); bfmalmeida@yahoo.com.br (B.F.M.d.A.); 3Laboratory of Veterinary Parasitology and Parasitic Diseases, Department of Veterinary Medicine, Federal University of Lavras, Lavras 37203-202, MG, Brazil; ld.barros@unesp.br; 4Department of Biomedical Science, Ross University School of Veterinary Medicine, Basseterre P.O. Box 334, Saint Kitts and Nevis; dbergfelt@rossvet.edu.kn; 5Departament of Clinics, Surgery and Animal Reproduction, Faculty of Veterinary Medicine of Araçatuba (FMVA), São Paulo State University (UNESP), Araçatuba 16050-680, SP, Brazil

**Keywords:** NSAID side effects, firocoxib safety, long-term treatment, leukogram, erythrogram, platelets

## Abstract

**Simple Summary:**

Non-steroidal anti-inflammatory drugs (NSAIDs) represent one of the most commonly used classes of drugs in both human and veterinary medicine. However, many clinical side effects have been observed, especially when treatment has been prolonged. While the anti-inflammatory efficacy and safety of repeated administration of firocoxib (Previcox^®^), which is a selective NSAID COX-2 inhibitor, has been evaluated for short-term use (one to fourteen days), its clinical relevance for longer-term use is not known. As a preliminary study, healthy, adult male and female horses (n = 7) were treated with daily oral firocoxib (0.11–0.14 mg/kg) for 40 days concomitant with the collection of blood samples encompassing treatment to assess hematological and biochemical endpoints. While this preliminary or pilot study was not without inherent limitations, treatment resulted in some hematological or biochemical effects that returned to pre-treatment or baseline values post-treatment without any clinically relevant adversity. Thus, the apparent favorable outcomes with the extended use of firocoxib provide a basis for future studies to reproduce these results with other more comprehensive study designs that may involve prolonged administration to healthy as well as injured or diseased horses to evaluate efficacy and the use of other clinically relevant endpoints.

**Abstract:**

Non-steroidal anti-inflammatory drugs (NSAIDs) represent one of the most commonly used classes of drugs in both human and veterinary medicine. However, many clinical side effects have been observed, especially when treatment has been prolonged. While the anti-inflammatory efficacy and safety of repeated administration of firocoxib (Previcox^®^), which is a selective NSAID COX-2 inhibitor, has been evaluated for short-term use (one to fourteen days), its clinical relevance for longer-term use is not known. As a preliminary study, healthy, adult male and female horses (n = 7) were treated with firocoxib for 40 days concomitant with the collection of blood samples encompassing treatment to assess hematological and biochemical endpoints. Daily oral administration of firocoxib was performed with one 57 mg tablet/animal (0.11–0.14 mg/kg), which was crushed and mixed with feed. Blood samples were collected one day before treatment (D0 or basal sample), during (D10, D20, D30, and D40), and after treatment (D55 and D70). Results indicated some hematological and biochemical effects were significantly reduced (*p* < 0.05) towards the end of treatment on D40 relative to pre-treatment or baseline values on D0. Post-treatment, all values returned to pre-treatment values within 30 days without any apparent clinical adversities. In conclusion, while these preliminary results are favorable for prolonged use of firocoxib in horses, future studies are required to evaluate the efficacy of prolonged use accompanied with other clinically relevant endpoints in healthy as well as injured or diseased animals.

## 1. Introduction

Non-steroidal anti-inflammatory drugs (NSAIDs) represent one of the most commonly used classes of drugs in both human and veterinary medicine. Apart from their anti-inflammatory effects in horses, NSAIDs are widely used to manage acute and chronic pain associated with laminitis and osteoarthritis [1,2] and endotoxemic shock [3,4,5], as well as provide therapeutic relief for sepsis in neonates [1]. Firocoxib is a second-generation NSAID that directly inhibits COX-2 with a strong safety profile that acts in the control of inflammation and pain [2,6,7]. Firocoxib is the first coxib NSAID-approved drug for use in horses. In regard to reproduction, the effect of firocoxib resulted in decreased inflammation associated with placentitis [8] and increased ovulation, especially when used to mitigate post-breeding inflammation during the periovulatory period in mares [9]. Firocoxib has also been effective in neonatal foals who are more susceptible to the adverse effects of non-selective NSAIDs compared to adult horses [10,11]. Subsequent to daily administration of firocoxib to neonatal foals for 7 days i.v. (0.09 mg/kg) [12] or orally for 9 days (0.1 mg/kg) [1], no apparent adverse effects were detected. While the anti-inflammatory efficacy and safety of repeated administration of firocoxib has been evaluated with a range of doses (0.09 to 0.3 mg/kg) from one to fourteen days [1,4,6,13,14,15,16,17,18,19], efficacy and safety of firocoxib associated with longer-term treatment has apparently not been documented in horses.

Classification of NSAIDs is conducted in accordance with their degree of selectivity for COX-1 or COX-2 enzymatic activity. The action of NSAIDs is primarily via inhibition of cyclooxygenase (COX) activity, which prevents the conversion of arachidonic acid to pro-inflammatory chemical mediators. While enzymatic isoform COX-1 is constitutively expressed and has been associated with functional aspects of the enterogastric mucosa, renal blood flow, vascular hemostasis, bone repair, myometrial contractions, and ovulation [1,4,7,20,21,22,23,24,25,26,27,28,29,30,31,32], enzymatic isoform COX-2 has been associated with regulation of the inflammatory response and formation of pro-inflammatory prostanoids [32].

While non-selective NSAIDs, such as phenylbutazone and flunixin meglumine, are effective anti-inflammatory medications, selective inhibition of COX-1 induces adverse side effects, such as renal damage, gastric ulcers, and colitis [14]. Conversely, the severity of side effects associated with the administration of highly selective COX-2 inhibitors, such as firocoxib to treat gastroenteritis in horses [2,4,5,13,14], is reduced. Consequently, however, if there is prolonged inflammation or ulceration in the enterogastric tract, inhibition of COX-2 can potentially lead to a delay in ulcer healing and exacerbation of inflammation [22].

Selective or non-selective NSAIDs apparently have no direct action on the lipoxygenase (LOX) pathway, which might show a limited role in the inflammatory cascade in the body when compared to steroidal anti-inflammatory drugs (SAIDs). Arachidonic acid is the most crucial precursor of inflammatory COX and LOX pathways. Leukotrienes are synthesized by LOX and play a significant role in the inflammation process. Since inhibition of COX enzymes (COX-1 and COX-2) stimulates the LOX pathway and increases leukotriene production, there is the possibility that long-term use of selective NSAIDs might increase the risk of cardiovascular and gastrointestinal diseases [33]. On the other hand, it is well known that continuous and long-term use of SAIDs is associated with gastrointestinal disease, suppression of immune function, laminitis, abortion, and premature foaling in pregnant mares and other unhealthy conditions [34].

As a preliminary or pilot study, the main objective was to assess the relevant safety of daily, long-term treatment (40 days) with firocoxib in healthy, adult male and female horses based on the evaluation of various hematological and biochemical endpoints encompassing the collection of blood samples before, during, and after treatment. It was hypothesized that long-term use of firocoxib would have little to nil clinical adversity in healthy horses and, therefore, support a more comprehensive efficacy study with injured or diseased horses.

## 2. Material and Methods

### 2.1. Animals, Treatment, and Blood Collection

This study was conducted in accordance with the principles of ethical animal experimentation, which was approved by the Board of Ethics in Animal Experimentation at CEUA-UniFIO (Protocol n. 010/2019; University Center of the Integrated Faculties of Ourinhos—UniFIO, Ourinhos, SP, Brazil).

A statistically sufficient sample size of seven healthy crossbred horses (six females and one male), 5–15 years of age, weighing 400–500 kg, in good body condition, and no clinical signs of disease or discomfort were involved in this study. Horses were housed at UniFIO School of Veterinary Medicine and maintained in paddocks with shelter, hay (*Cynodon dactylon*), concentrates, mineral supplements, and free access to water.

All animals were treated orally with 0.11–0.14 mg/kg (57 mg tablet/animal) of firocoxib (Previcox^®^, Merial, Campinas, SP, Brazil) once a day for 40 days. Each tablet was crushed and placed on top of feed concentrate provided in individual troughs. Oral administration of firocoxib was considered less invasive, safer, and more practical. Each animal was closely observed during feeding to ensure that all the feed and firocoxib were consumed. Blood samples were collected from the jugular vein one day before the beginning of treatment (D0 or basal sample), during treatment on D10, D20, D30, and D40, and after treatment on D55 and D70. Blood samples were collected into two different vacutainer tubes, one without and one with anticoagulant (EDTA) for hematological and biochemical analysis, respectively. All samples were sent immediately to the lab at Roque Quagliato Veterinary Hospital, UniFIO, for single-sample analysis.

Horses were monitored daily for clinical signs of abdominal pain, attitude, behavior, food and water intake, oral ulcers, fecal consistency, and body condition from D0 to D70.

### 2.2. Hematological and Biochemical Analyses

Various hematological and biochemical endpoints were determined by single-sample analysis. Hematological analyses of red blood cells, leukocytes, platelet count, and hemoglobin were performed by a veterinary automated cell counter (ABX Micros ESV 60, Paris, France). Packed cell volume (PCV) was determined by the Strumia microcapillary method (11,400 rpm for 5 min), and the differential count of leukocytes together with the morphological evaluation of red blood cells, leukocytes, and platelets were performed on blood smears stained with commercial hematological dye (Instant-Prov, Newprov, Pinhais, PR, Brazil).

Biochemical analysis of serum was subjected to a semi-automated photocolorimeter (BIO 2000, BioPlus, Barueri, SP, Brazil) using commercial reagents (Labtest Diagnóstica SA, Lagoa Santa, MG, Brazil) according to the manufacturer’s recommendation. Biochemical determinations were carried out in duplicate at 37 °C after calibration (Calibrator Calibra H, Labtest Diagnóstica SA, Lagoa Santa, MG, Brazil) and verification with level I (Qualitrol 1H, Labtest Diagnóstica SA, Lagoa Santa, MG, Brazil) and II (Qualitrol 2H, Labtest Diagnóstica SA, Lagoa Santa, MG, Brazil) commercial controls.

Aspartate aminotransferase (AST) activity was quantitated by ultraviolet kinetic methodology, creatinine by alkaline picrate colorimetric method—Jaffé, alkaline phosphatase (ALP) by Bowers kinetic method and modified McComb, gamma-glutamyltransferase (GGT) by modified Szasz method, total plasma protein by the colorimetric method of biuret, and urea according to UV enzymatic methodology.

Hematological results were evaluated and referenced based on those reported by Thrall (2007) [35], and biochemical and jaundice index results were evaluated and referenced based on those by Kaneko et al. (2008) [36] and Robinson and Sprayberry (2003) [34].

### 2.3. Statistical Analysis

An estimate of sample size was conducted using R software in the RStudio integrated development environment (version 4.1.0) with the “power *t*-test” function from the “stats” package, setting 0.80 power, and 0.05 as the significance level for the two-tailed paired sample hypothesis test. For the AST variable, the minimum difference to be detected (306) was achieved by subtracting the value found in the control group (411) and the group treated with phenylbutazone (717) when the minimum difference was detected, and the standard deviation found in this group (40) was obtained from a previous study carried out in equids [37].

All data were evaluated using the Graph Pad Prism 6 and Epilnfo 7 programs. The Shapiro–Wilk test was used to assess the normality of WBC and biochemical variables. Data with normal distribution were represented as mean (±SEM), and those data not normally distributed were represented as median (min–max). Analysis of variance (ANOVA) followed by Dunnet of Friedman + Dunn method was used to assess statistical differences before treatment on D0 and subsequent times during and after treatment. A 5% significance level was used to determine statistical differences.

Erythrogram data were presented as mean (±SEM) or median (min–max) and changes of clinical importance were discussed based on reference data for the species.

## 3. Results

Hematological and biochemical results before treatment (D0), during (D10, D20, D30, and D40), and after (D55 and D70) treatment, inclusive of statistical results, are shown in Table 1, Table 2 and Table 3. For clarity of individual and group responses to treatment, the percentage of animals with values under, within, and above reference limits at each moment are also indicated.

In Table 1, mean values were lower (*p* < 0.05) for erythrocytes, hemoglobin, PCV, and RDW during treatment (D30 and D40) and after treatment (D55) but not at 30 days post-treatment (D70). In Table 2, mean values in total leukocytes and plasmatic proteins were lower (*p* < 0.05) during (D40) and after (D55) treatment but not at 30 days post-treatment (D70). In Table 3, mean values were lower (*p* < 0.05) for AST (D20, D30, D40), ALP (D10, D20, D40), and creatinine (D10-40). Post-treatment values were still lower for AST and ALP on D55 but not on D70, and creatinine remained lower on D55 and D70. The jaundice index did not change throughout the experiment.

Horses did not show any clinical signs of NSAID gastroenteric toxicity, such as abdominal pain, oral ulcers, reduced food intake, or diarrhea, during (D0 to D40) and after (D55 to D70) treatment.

## 4. Discussion

Oral administration of the selective COX-2 NSAID firocoxib to healthy adult horses for 40 days indicated that prolonged treatment significantly affected some hematological and biochemical endpoints during treatment relative to baseline values before treatment, but mean values, except for creatinine, returned to pre-treatment values within 30 days post-treatment without detection of any clinically relevant adversity. Although not necessarily of clinical relevance, erythrocyte, lymphocytes, total leucocytes, and ALP values were below reference in 14% (1/7), 14% (1/7), 14% (1/7), and 57% (4/7) of animals, respectively. Additionally, horses did not show any clinical signs of NSAID gastroenteric toxicity during (D0 to D40) or after (D55 to D70) treatment.

Potential adverse effects associated with NSAIDs other than firocoxib have included gastrointestinal and renal toxicity [38,39]. Specifically, while a seven-day treatment regimen with the non-selective NSAID phenylbutazone was associated with gastric ulceration in horses [40], treatment with firocoxib, which has been shown to be equivalent to phenylbutazone, was clinically efficacious without adversity [2]. Correspondingly, oral administration of firocoxib for 14 days in horses resulted in reduced lameness scores without any adverse side effects [2,41]. Thus, while some hematological and biochemical results reflective of side effects were detected, the combined results from previous studies with firocoxib for 14 days [2,6,41] and present results for 40 days indicated most, if not all, effects were reversible. Additionally, no clinically adverse signs were evident after 40 days of treatment with firocoxib, while previous studies [37,42] with non-selective NSAIDs (e.g., phenylbutazone, flunixin meglumine, and ketoprofen) have shown mild to severe anorexia, diarrhea, oral ulcers, and colic signs within 12 days of treatment.

In a previous study [37] using other non-selective NSAIDs, mean values of erythrocytes, hemoglobin, and PCV were lower within 12 days of treatment, which occurred much earlier than what was detected in the present study on D30 and D40 vs D0 with firocoxib. In the present study, a reduction in hematological parameters caused anemia in two of the treated horses, which was accompanied by total protein reduction; however, values returned to pre-treatment values by D70 or 30 days after treatment. While the basis of these observations is unknown, perhaps there was some hemorrhage or increase in plasmatic volume. In previous studies that have evaluated the adverse effects of non-selective NSAIDs on donkeys [37] and horses [42], one side effect was observed without any detectable presence of fecal occult blood. Thus, further investigations are needed to determine the physiopathology of these changes during treatment with non-selective NSAIDs.

While the total leukocyte value was lower on the last day of treatment (D40), values returned to pre-treatment values within 30 days post-treatment (D70). Neutropenia has been associated with the long-term use of other NSAIDs [43]. In the present study, even though a statistically significant reduction in neutrophil count was not detected, mean values for different lines of leukocytes were slightly reduced. Only one horse showed lymphocyte counts 14% below reference limits on the last day of treatment (D40) and for 30 days thereafter (D70). Overall, while some hematological values were reduced later during treatment, there was a return to pre-treatment values within 30 days post-treatment with no observed clinical adversity.

Relative to pre-treatment (D0), the mean total plasmatic protein (TPP) value was lower on the last day of treatment (D40) and post-treatment (D55) but returned to pre-treatment values by 30 days (D70). While the sum of albumin, fibrinogen, and globulin levels defines TPP, a reduction in globulins has been the basis for the reduced values of TPP. The basis for the lower TPP values in the present study is unknown but has been related to reduced globulin or hypoalbuminemia [35] since no weight loss or anorexia was observed in the animals during the study to justify malnutrition. Correspondingly, reduced globulins have been associated with renal and gastrointestinal bleeding [44], which can result from NSAID treatment [13]. The hypoproteinemia observed in this study could also be associated with hypoalbuminemia, as seen in other NSAIDs [42]; however, a limitation of the present study was the lack of information on albumin concentration, which is a routine aspect of biochemical monitoring and critically important in the assessment of hypoproteinemia, which was one of the significant abnormalities identified in this study. A future study is required for clarification.

A commonly reported side effect of NSAIDs is their hepatic [45] and renal [46] toxicity. While the pharmacokinetics and hepatic and renal metabolic mechanisms of actions of NSAIDs are not fully known, hepatic metabolism results in inactive metabolites that are excreted in bile and urine [47]. In the present study, AST, ALP, and GGT enzymatic concentrations were used to evaluate liver function. An increase in GGT and ALP concentrations is associated with obstructive hepatic disease and, with a narrow reference limit for GGT, it is a more sensitive diagnostic indicator for cholestasis than ALP [35]. Relative to pre-treatment values, there were no changes in GGT, AST was lower on D20, D30, and D40, and ALP was lower on D10, D20, and D40 during treatment. Post-treatment, concentrations of AST and ALP increased on D70 or 15 or 30 days after treatment. Even though there may be other more liver-specific enzymes indicative of hepatocellular injury and cholestasis, the present results suggested the absence of major liver toxicity. While creatinine showed a slight reduction in 29% to 43% of the animals, levels went below reference limits during treatment (D10 to D40) without any detectable changes in urea. Moreover, after the end of treatment, creatinine returned to pre-treatment values. Low creatinine is potentially associated with weight loss and muscle loss [35,36], but no changes in appetite and weight were detected during the experiment. Thus, while there were temporary reductions in liver enzymes and kidney-related creatinine concentrations, these transient changes were considered clinically irrelevant.

In summary, prolonged treatment with firocoxib caused changes in erythrocytes, hemoglobin, PCV, leukocytes, TP, AST, ALP, and creatinine when compared to basal levels. However, values were still within respective reference ranges; hence, there were no clinically significant changes, and decreased liver enzymatic activity (AST and ALP) and creatinine were not necessarily meaningful. While this preliminary or pilot study was not without inherent limitations, daily oral administration of firocoxib (0.11–0.14 mg/kg) as a selective NSAID COX-2 inhibitor for 40 days in healthy adult male and female horses resulted in some hematological or biochemical effects during treatment that returned to pre-treatment or baseline values post-treatment without any clinically relevant adversity. Thus, the apparent favorable outcomes with the extended use of firocoxib provide a basis for future studies to reproduce these results with other more comprehensive study designs that may involve prolonged administration to healthy as well as injured or diseased horses to evaluate efficacy and use of other clinically relevant endpoints.

## Figures and Tables

**Table 1 vetsci-11-00256-t001:** Mean (±SEM) values and percentage of animals below, within, and above reference limits of various hematological endpoints in adult horses (n = 7) treated with firocoxib (0.11 mg/kg) orally once a day for 40 days from before treatment (D0 or baseline), during treatment (D10, D20, D30, and D40), and after treatment (D55 and D70).

Endpoints	Before	During Treatment				After		Reference
	D0	D10	D20	D30	D40	D55	D70	
Erythrocyte (10^12^/L)	8.71 (±3.31)	8.11 (±0.32)	8.32 (±0.40)	7.75 (±0.34)	7.11 * (±0.39)	7.17 * (±0.26)	8.14 (±0.36)	6.5 to 12.5
Below	0% (0/7)	0% (0/7)	0% (0/7)	14% (1/7)	14% (1/7)	14% (1/7)	14% (1/7)	
Within	100% (7/7)	100% (7/7)	100% (7/7)	86% (6/7)	86% (6/7)	86% (6/7)	86% (6/7)	
Above	0% (0/7)	0% (0/7)	0% (0/7)	0% (0/7)	0% (0/7)	0% (0/7)	0% (0/7)	
Hemoglobin (g/dL)	13.87 (±0.45)	13.06 (±0.38)	13.44 (±0.63)	12.07 * (±0.47)	11.53 * (±0.61)	11.74 * (±0.37)	13.66 (±0.54)	11 to 18
Below	0% (0/7)	0% (0/7)	0% (0/7)	29% (2/7)	43% (3/7)	29% (2/7)	0% (0/7)	
Within	100% (7/7)	100% (7/7)	100% (7/7)	71% (5/7)	57% (4/7)	71% (5/7)	100% (7/7)	
Above	0% (0/7)	0% (0/7)	0% (0/7)	0% (0/7)	0% (0/7)	0% (0/7)	0% (0/7)	
PCV (%)	41.43 (±1.63)	38 (±1.04)	39.14 (±1.83)	35.71 * (±1.53)	33.29 * (±1.84)	34.14 * (±1.16)	39.29 (±1.62)	32 to 52
Below	0% (0/7)	0% (0/7)	14% (1/7)	29% (2/7)	43% (3/7)	29% (2/7)	0% (0/7)	
Within	100% (7/7)	100% (7/7)	86% (6/7)	71% (5/7)	57% (4/7)	71% (5/7)	100% (7/7)	
Above	0% (0/7)	0% (0/7)	0% (0/7)	0% (0/7)	0% (0/7)	0% (0/7)	0% (0/7)	
MCV (f/L)	46.86 (±1.1)	46.43 (±1.02)	46.86 (±1.12)	46.57 (±1.15)	47.29 (±1.14)	47.71 (±1.19)	48.43 (±1.15)	39 to 50
Below	0% (0/7)	0% (0/7)	0% (0/7)	0% (0/7)	0% (0/7)	0% (7/7)	0% (7/7)	
Within	100% (7/7)	100% (7/7)	100% (7/7)	100% (7/7)	100% (7/7)	86% (6/7)	86% (6/7)	
Above	0% (0/7)	0% (0/7)	0% (0/7)	0% (0/7)	0% (0/7)	14% (1/7)	14% (1/7)	
MCHC (g/dL)	34.11 (±0.19)	34.71 (±0.11)	34.33 (±0.33)	33.71 (±0.18)	34.49 (±0.15)	34.37 (±0.15)	34.79 (±0.11)	34 to 39
Below	--	0% (0/7)	29% (2/7)	86% (6/7)	14% (1/7)	29% (2/7)	0% (0/7)	
Within	--	100% (7/7)	71% (5/7)	14% (1/7)	86% (6/7)	71% (5/7)	100% (7/7)	
Above	--	0% (0/7)	0% (0/7)	0% (0/7)	0% (0/7)	0% (0/7)	0% (0/7)	
RDW (%)	19.79 (±0.18)	19.76 (±0.22)	20.27 (±0.26)	20.51 * (±0.19)	20.64 * (±0.22)	20.66 * (±0.22)	20.37 (±0.15)	-

PCV—packed cell volume; MCV—mean cell volume; MCHC—mean cell hemoglobin concentration; * Significantly different from baseline (*p* < 0.05); Reference values for horses based on THRALL, 2007 [35].

**Table 2 vetsci-11-00256-t002:** Mean (±SEM), median (min–max), and percentage of animals below, within, and above reference limits of various biochemical endpoints with and without normal distribution in adult horses (n = 7) treated with firocoxib (0.11 mg/kg) orally once a day for 40 days from before treatment (D0 or baseline), during treatment (D10, D20, D30, and D40), and after treatment (D55 and D70).

Endpoints	Before	During Treatment				After		Reference
	D0	D10	D20	D30	D40	D55	D70	
Band cells (µL)	0 (0–0)	0 (0–0)	0 (0–0)	0 (0–0)	0 (0–0)	0 (0–0)	0 (0–0)	0 to 100
Below	0% (0/7)	0% (0/7)	0% (0/7)	0% (0/7)	0% (0/7)	0% (0/7)	0% (0/7)	
Within	100% (7/7)	100% (7/7)	100% (7/7)	100% (7/7)	100% (7/7)	100% (7/7)	100% (7/7)	
Above	0% (0/7)	0% (0/7)	0% (0/7)	0% (0/7)	0% (0/7)	0% (0/7)	0% (0/7)	
Segmented (µL)	4570.7 (±304.1)	3736.9 (±298.8)	4650.6 (±232.2)	4093.4 (±328.5)	3784.3 (±195.8)	3549.1 (±183.3)	3979.4 (±548.4)	2700 to 6700
Below	0% (0/7)	0% (0/7)	0% (0/7)	0% (0/7)	0% (0/7)	0% (0/7)	0% (0/7)	
Within	100% (7/7)	100% (7/7)	100% (7/7)	100% (7/7)	100% (7/7)	100% (7/7)	100% (7/7)	
Above	0% (0/7)	0% (0/7)	0% (0/7)	0% (0/7)	0% (0/7)	0% (0/7)	0% (0/7)	
Monocytes (µL)	376 (164–546)	370 (75–504)	152 (89–780)	156 (89–780)	280 (134–608)	268 (44–462)	222 (66–576)	0 to 800
Below	0% (0/7)	0% (0/7)	0% (0/7)	0% (0/7)	0% (0/7)	0% (0/7)	0% (0/7)	
Within	100% (7/7)	100% (7/7)	100% (7/7)	100% (7/7)	100% (7/7)	100% (7/7)	100% (7/7)	
Above	0% (0/7)	0% (0/7)	0% (0/7)	0% (0/7)	0% (0/7)	0% (0/7)	0% (0/7)	
Lymphocytes (µL)	2743.4 (±327.4)	2838.7 (±387.2)	2844.3 (±427.8)	2710.9 (±263.2)	2320.6 (±309.7)	2560.7 (±272.2)	2570 (±364.2)	1500 to 5500
Below	0% (0/7)	14% (1/7)	0% (0/7)	0% (0/7)	14% (1/7)	14% (1/7)	14% (1/7)	
Within	100% (7/7)	86% (6/7)	100% (7/7)	100% (7/7)	86% (6/7)	86% (6/7)	86% (6/7)	
Above	0% (0/7)	0% (0/7)	0% (0/7)	0% (0/7)	0% (0/7)	0% (0/7)	0% (0/7)	
Eosinophil (µL)	204.3 (±40.3)	169.3 (±44.2)	264.7 (±94.7)	342.3 (±85.9)	223.4 (±31.8)	195.7 (±46.9)	269.9 (±91.4)	0 to 900
Below	0% (0/7)	0% (0/7)	0% (0/7)	0% (0/7)	0% (0/7)	0% (0/7)	0% (0/7)	
Within	100% (7/7)	100% (7/7)	100% (7/7)	100% (7/7)	100% (7/7)	100% (7/7)	100% (7/7)	
Above	0% (0/7)	0% (0/7)	0% (0/7)	0% (0/7)	0% (0/7)	0% (0/7)	0% (0/7)	
Basophil (µL)	63 (0–188)	0 (0–150)	0 (0–0)	0 (0–81)	74 (0–198)	0 (0–156)	0 (0–150)	0 to 200
Below	0% (0/7)	0% (0/7)	0% (0/7)	0% (0/7)	0% (0/7)	0% (0/7)	0% (0/7)	
Within	100% (7/7)	100% (7/7)	100% (7/7)	100% (7/7)	100% (7/7)	100% (7/7)	100% (7/7)	
Above	0% (0/7)	0% (0/7)	0% (0/7)	0% (0/7)	0% (0/7)	0% (0/7)	0% (0/7)	
Total Leucocytes (µL)	7942.9 (±353.8)	7200 (±316.2)	8014.3 (±516.1)	7385.7 (±458.9)	6714.3 * (±346.7)	6614.3 * (±437.2)	7100 (±537.6)	5500 to 12,000
Below	0% (0/7)	0% (0/7)	0% (0/7)	14% (1/7)	14% (1/7)	14% (1/7)	14% (1/7)	
Within	100% (7/7)	100% (7/7)	100% (7/7)	86% (6/7)	86% (6/7)	86% (6/7)	86% (6/7)	
Above	0% (0/7)	0% (0/7)	0% (0/7)	0% (7/7)	0% (7/7)	0% (7/7)	0% (7/7)	
Platelets (×10^9^/L)	214.6 (±9.5)	221.9 (±14.3)	244.1 (±21.0)	228.1 (±12.5)	269.1 (±18.2)	216.4 (±13.5)	219.3 (13.5)	150 to 500
Below	0% (0/7)	0% (0/7)	0% (0/7)	0% (0/7)	0% (0/7)	0% (0/7)	0% (0/7)	
Within	100% (7/7)	100% (7/7)	100% (7/7)	100% (7/7)	100% (7/7)	100% (7/7)	100% (7/7)	
Above	0% (0/7)	0% (0/7)	0% (0/7)	0% (0/7)	0% (0/7)	0% (0/7)	0% (0/7)	
Total plasmatic protein (g/dL)	6.8 (6.6–7)	6.7 (6.2–6.6)	7 (6.6–7.8)	6.2 (6–6.6)	6 * (5.8–6.4)	6 * (6–6.2)	6.8 (6.6–7)	6.0 to 8.0
Below	0% (0/7)	0% (0/7)	0% (0/7)	0% (0/7)	0% (0/7)	14% (1/7)	0% (0/7)	
Within	100% (7/7)	100% (7/7)	100% (7/7)	100% (7/7)	100% (7/7)	86% (6/7)	100% (7/7)	
Above	0% (0/7)	0% (0/7)	0% (0/7)	0% (0/7)	0% (0/7)	0% (7/7)	0% (0/7)	
Fibrinogen (g/dL)	0.2 (0.2–0.4)	0.2 (0.2–0.4)	0.2 (0.2–0.6)	0.2 (0.2–0.6)	0.2 (0.2–0.4)	0.2 (0.2–0.2)	0.2 (0.2–0.4)	0.1 to 0.4
Below	0% (0/7)	0% (0/7)	0% (0/7)	0% (0/7)	0% (0/7)	0% (0/7)	0% (0/7)	
Within	86% (6/7)	100% (7/7)	86% (6/7)	86% (6/7)	86% (6/7)	100% (7/7)	100% (7/7)	
Above	14% (1/7)	0% (0/7)	14% (1/7)	14% (1/7)	14% (1/7)	0% (0/7)	0% (0/7)	

* Significantly different from baseline (*p* < 0.05); Reference values for horses based on THRALL, 2007 [35].

**Table 3 vetsci-11-00256-t003:** Mean (±SEM) values and percentage of animals below, within, and above reference limits of hepatic enzymes, creatinine, and urea in adult horses (n = 7) treated with firocoxib (0.11 mg/kg) orally once a day for 40 days from before treatment (D0 or baseline), during treatment (D10, D20, D30, and D40), and after treatment (D55 and D70).

Endpoints	Before	During Treatment				After		Reference
	D0	D10	D20	D30	D40	D55	D70	
AST (U/L)	341.42 (±12.05)	245.71 (±39.43)	242 * (±23.66)	222.42 * (±22.58)	180.85 * (±20.37)	292.71 * (±12.37)	318.85 (±16.35)	226 to 366
Below	0% (0/7)	29% (2/7)	29% (2/7)	29% (2/7)	86% (6/7)	0% (0/7)	0% (0/7)	
Within	86% (6/7)	57% (4/7)	71% (5/7)	71% (5/7)	14% (1/7)	100% (7/7)	86% (6/7)	
Above	14% (1/7)	14% (1/7)	0% (0/7)	0% (0/7)	0% (0/7)	0% (0/7)	14% (1/7)	
ALP (U/L)	159.57 (±16.57)	129.85 * (±11.33)	120.28 * (±8.26)	260.14 (±45.41)	119.14 * (±21.51)	118.14 * (±9.40)	139.42 (±10.14)	143 to 395
Below	29% (2/7)	57% (4/7)	86% (6/7)	14% (1/7)	29% (2/7)	86% (6/7)	57% (4/7)	
Within	71% (5/7)	43% (3/7)	14% (1/7)	86% (6/7)	71% (5/7)	14% (1/7)	43% (3/7)	
Above	0% (0/7)	0% (0/7)	0% (0/7)	0% (0/7)	0% (0/7)	0% (0/7)	0% (0/7)	
GGT (U/L)	15 (0)	26.71 (±2.25)	20.42 (±3.34)	24.42 (±3.34)	20.14 (±2.10)	23.42 (±3.10)	20.14 (2.10)	4 to 44
Below	0% (0/7)	0% (0/7)	0% (0/7)	0% (0/7)	0% (0/7)	0% (0/7)	0% (0/7)	
Within	100% (7/7)	100% (7/7)	100% (7/7)	100% (7/7)	100% (7/7)	100% (7/7)	100% (7/7)	
Above	0% (0/7)	0% (0/7)	0% (0/7)	0% (0/7)	0% (0/7)	0% (0/7)	0% (0/7)	
Creatinine (mg/dL)	1.77 (±0.07)	1.3 * (±0.05)	1.28 * (±0.10)	1.28 * (±0.16)	1.2 * (±0.16)	1.6 * (±0.07)	1.67 * (±0.07)	1.2 to 1.9
Below	0% (0/7)	29% (2/7)	43% (3/7)	43% (3/7)	43% (3/7)	0% (0/7)	0% (0/7)	
Within	100% (7/7)	71% (5/7)	57% (4/7)	57% (4/7)	43% (3/7)	100% (7/7)	100% (7/7)	
Above	0% (0/7)	0% (0/7)	0% (0/7)	0% (0/7)	14% (1/7)	0% (0/7)	0% (0/7)	
Urea (mg/dL)	34.14 (±2.38)	37.42 (±1.73)	42 (±3.47)	32.42 (±2.38)	31.14 (±3.02)	32.14 (±1.20)	34 (±1.32)	21 to 51
Below	0% (0/7)	0% (0/7)	0% (0/7)	0% (0/7)	0% (0/7)	0% (0/7)	0% (0/7)	
Within	100% (7/7)	100% (7/7)	86% (6/7)	100% (7/7)	100% (7/7)	100% (7/7)	100% (7/7)	
Above	0% (0/7)	0% (0/7)	14% (1/7)	0% (0/7)	0% (0/7)	0% (0/7)	0% (0/7)	

AST—aspartate aminotransferase; ALP—alkaline phosphatase; GGT—gamma-glutamyltransferase; * Significantly different from baseline (*p* < 0.05); Reference values for horses based on Kaneko et al., 2008 [36], and Robinson and Sprayberry, 2003 [34].

## Data Availability

Data are contained within the article.

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
