# Peer review of "Hematological and Biochemical Effects Associated with Prolonged Administration of the NSAID Firocoxib in Adult Healthy Horses"

_vetsci, 2024, doi:10.3390/vetsci11060256_

Round 1
Reviewer 1 Report
Comments and Suggestions for Authors
This study provides a very topline view of the hematological and biochemical effects of firocoxib following 40 days of treatment.
1. The authors should address why they didn’t do endoscopic examinations the horses to access gastric ulcer formation prior to or at the conclusion of the study.
2. The authors state on line 173 value returned to normal for “most” end-points, in fact only one end-point (creatinine) didn’t return to the pre-treatment values.
3. The authors include any observations made regarding the occurrence of diarrhea, oral ulcers or signs of colic, as described in Part 4 Discussion, for studies conducted with other NSAIDs.
Author Response
This study provides a very topline view of the hematological and biochemical effects of firocoxib following 40 days of treatment.
1. The authors should address why they didn’t do endoscopic examinations the horses to access gastric ulcer formation prior to or at the conclusion of the study.
- Agreed, endoscopic examinations would have give great information about gastric sanity during treatment; however, we did not have a endoscopy at the UniFIO farm that we could use for this purpose not even enough resources to pay for it.
- The authors state on line 173 value returned to normal for “most” end-points, in fact only one end-point (creatinine) didn’t return to the pre-treatment values.
- Changes were made in the text to be more specific about what happened (line 209, first paragraph of discussion section), as shown below:
“Oral administration of the selective COX-2 NSAID firocoxib to healthy adult horses for 40 days indicated that prolonged treatment significantly affected some hematological and biochemical endpoints during treatment relative to baseline values before treatment but mean values, except for creatinine, returned to pre-treatment values within 30 days post-treatment without detection of any clinically relevant adversity. Although not necessarily of clinical relevance, erythrocyte, lymphocytes, total leucocytes, and ALP values were below reference in 14% (1/7), 14% (1/7), 14% (1/7) and 57% (4,7) of animals, respectively”
- The authors include any observations made regarding the occurrence of diarrhea, oral ulcers or signs of colic, as described in Part 4 Discussion, for studies conducted with other NSAIDs.
- Changes were made in the text (line 212, first paragraph of Discussion section).
“Additionally, horses did not show any clinical signs of NSAIDs gastroenteric toxicity during (D0 to D40) or after (D55 to D70) treatment..”

Reviewer 2 Report
Comments and Suggestions for Authors
Dear author,
this paper describes the changes in the most commonly used haematological and hepatic/renal biochemical parameters in seven horses during 40 days of treatment with oral firocoxib. The title appropriately describes the study: "Haematological and Biochemical Effects Associated with Prolonged Administration etc...etc...". However, in the text of the manuscript, the authors in many places suggest to the reader that this article can say something about the safety of long-term administration of firocoxib to the horse. Moderate increases or decreases in hepatic/renal parameters say little about the safety of long-term administration of an NSAID. If the authors had performed urinalysis, coagulation tests (PT; APTT; viscoelastic coagulation parameters); fecal occult blood test; fundoscopic examination of the eye; gastro and duodenal endoscopy, possibly with intestinal mucosal biopsies, this study would have had more to say about long-term effects and safety.
In my opinion, the data reported in this study can be summarised in a short communication where it will be important to emphasise that it can say little (nothing?) about the safety of prolonged administration of firocoxib in horses.
Best regards
Author Response
regards
- The authors appreciate and agreed with the comments and suggestion of the reviewer, however, this preliminary or pilot study was not designed to fully evaluate the safety of prolonged administration of firocoxib in horses. Behavioral and physiological endpoints were monitored and assessed encompassing before, during, and after treatment in healthy horses to determine any overt adversity during its extended use. The results indicated little if any clinically relevant adversity, which supports the extended use of firocoxib in horses but, moreover, provides a basis for a more comprehensive study to evaluate the efficacy of the extended use of the medication in injured or diseased horses. All of this has been clarified in respective sections in the revised version of this manuscript.
Reviewer 3 Report
Comments and Suggestions for Authors
The reviewer would like to thank the authors for the submission of the paper titled: Hematological and biochemical effects associated with prolonged administration of the NSAID firocoxib in adult healthy horses. Please find below the suggested revisions to make the manuscript better for publication.
There are four areas in this manuscript that need addressed to make the manuscript much stronger for publication.
1. When selecting the animals for the study you list that there were 7 horses that were chosen. Please provide a power calculation for how you decided that 7 horses would provide enough variance/difference if there was to be statistically significance. If this paper was only to be descriptive in nature, listing potential side effect outcomes then this must clearly be stated as the outcome. It is stated that this is a preliminary study, it may be better to classify this as a pilot project. Are there plans to continue this work in a broader hypothesis generated clinical trial, if so it may add significant weight to the manuscript if this is stated. This also may help with that there is no clear research hypothesis stated.
2. In the introduction can you also please clearly state that COX selective vs nonselective inhibitors have no direct known action on the lipoxygenase pathway and this may and likely does play a large role in the inflammatory cascade in the body. Please mention how this pathway works.
3. As far as the medication firicoxib how did you decide on 40 days of therapy, please provide justification for this timeframe. Also, please justify how you chose the blood draw time points that you did was this based on previous PK/PD data and known excretion data. You also need to justify for the use of oral medication vs intravenous formulation. How did you make sure that each animal received each dose each day. Stating that the medication was crushed up and placed in the feed is assuming that all of it was consumed. You need to provide data that it was absorbed into the horse and it was at a steady state concentration to state that the medication HAD or DID NOT have hematologic effects on the animal. You have know proof in the current form of the manuscript that the meds were clinically at therapeutic levels in the horse. Also, you need to account for size of the patient and how this may affect this.
4. Blood results and tables.
Routinely ALT is not measured in the horse and is not a reliable indicator of hepatic injury. It is not a viable enzymatic marker in the horse, please remove. Did you consider looking at liver functionality such as serum bile acids. This would add more validity to your manuscript as far as assessing functionality of liver mass in each horse. Also were these samples run only once or were they run in duplicate or triplicate to assess for variation. If they were not please state in the manuscript.
The tables are very hard to understand from a statistical significance because it does not match up to the reference range provided for each analyte. This was for erythrocytes, hemoglobin, PCV, leukocytes, TP, AST, ALP and creatinine. All of these while stated were outside of the SEM but were still within the stated reference range so how significant is this really. This needs to be addressed better in the discussion. Also, you have to consider for liver enzymatic activity when it decreases this clinically means nothing, this is also the same consideration for creatinine. Please revise this portion of the discussion.
Globally this manuscript just provides descriptive information about longterm use of firocoxib and potential hematological effects on a very small sample size. Please consider adding more animals or justification for the size of the sample pool. While there may be some clinical value in knowing that we may be able to leave horses on this medication for an extended period of time with minimal side effects in the current state of this manuscript and how it is presented that cannot be adequately determined without additional information and revision.
Author Response
The reviewer would like to thank the authors for the submission of the paper titled: Hematological and biochemical effects associated with prolonged administration of the NSAID firocoxib in adult healthy horses. Please find below the suggested revisions to make the manuscript better for publication.
There are four areas in this manuscript that need addressed to make the manuscript much stronger for publication.
- When selecting the animals for the study you list that there were 7 horses that were chosen. Please provide a power calculation for how you decided that 7 horses would provide enough variance/difference if there was to be statistically significance. If this paper was only to be descriptive in nature, listing potential side effect outcomes then this must clearly be stated as the outcome. It is stated that this is a preliminary study, it may be better to classify this as a pilot project. Are there plans to continue this work in a broader hypothesis generated clinical trial, if so it may add significant weight to the manuscript if this is stated. This also may help with that there is no clear research hypothesis stated.
- The sample size estimation was added to the statistical analysis in the Material and Methods section: “The analysis to estimate sample size was conducted by a biological data scientist in R software in the RStudio integrated development environment (version 4.1.0) using the “power.t.test” function from the “stats” package, setting 0.80 power and 0.05 as significance level for two-tailed paired samples hypothesis test. For the AST variable, the minimum difference to be detected (306) was achieved by subtracting the value found in the control group (411) and the group treated with phenylbutazone (717) when the minimum difference was detected and the standard deviation found in this group (40) obtained from a previous study carried out in equids [40]. According to the aforementioned limits, it is estimated that the sample size is at least 7 horses.”
It is also important to address the difficulties on working with horses and to having a large number of animals to work with, which can be also seen by the sample size used in the references used in this manuscript, for example:
Hovanessian et al., 2013 tested pharmacokinetic and side effects of firocoxib in 7 healthy neonatal foals
Macpherson et al., 2020 tested firocoxib effects in 6 mares with placentitis
Aranzales et al., 2014 evaluated the oral administration on phenylbutazone by using 15 horses that were divided in three groups with only 5 horses per group.
- In the introduction can you also please clearly state that COX selective vs nonselective inhibitors have no direct known action on the lipoxygenase pathway and this may and likely does play a large role in the inflammatory cascade in the body. Please mention how this pathway works.
- Agreed, and changes in the Introduction section have been done in order to include this informations: “Selective or non-selective NSAIDs have no direct action on the lipoxygenase (LOX) pathway which might show a limited role in the inflammatory cascade in the body when compared to steroidal anti-inflammatory drugs. Arachidonic acid is the most crucial precursor of inflammatory COX and LOX pathways. Leukotrienes are synthesized by LOX and play a significant role in the inflammation process. Since inhibition of COX enzymes (COX-1 and COX-2) stimulates the LOX pathway and increases leukotriene production, the use of selective NSAIDs might increase the risk of cardiovascular and gastrointestinal diseases [Mukhopadhyay et al., 2023]. On the other hand, the continuous and long use of SAIDs, are associated with gastrointestinal diseases, suppression of immune function, laminitis, abortion/premature foaling in pregnant mares and other [35].
The hypothesis is that firocoxib will be a safe choice for long anti-inflammatory treatment for horses causing none or little clinical alterations and, as a preliminary study, this work aims to evaluate the potential relevance for long-term use in horses, healthy, adult male and female horses were treated daily with firocoxib for 40 days, which was accompanied by collection of blood samples before, during and, after treatment to assess various hematological and biochemical endpoints.”
- As far as the medication firicoxib how did you decide on 40 days of therapy, please provide justification for this timeframe. Also, please justify how you chose the blood draw time points that you did was this based on previous PK/PD data and known excretion data. You also need to justify for the use of oral medication vs intravenous formulation. How did you make sure that each animal received each dose each day. Stating that the medication was crushed up and placed in the feed is assuming that all of it was consumed. You need to provide data that it was absorbed into the horse and it was at a steady state concentration to state that the medication HAD or DID NOT have hematologic effects on the animal. You have know proof in the current form of the manuscript that the meds were clinically at therapeutic levels in the horse. Also, you need to account for size of the patient and how this may affect this.
- This study was designed as a preliminary or pilot to evaluate the practical application of firocoxib and determine the duration in which it could be applied without adversity in health horses. This was not a PK/PD study. This aspect will be considered in a future study to address the efficacy of prolonged treatment with firocoxib in injured or diseases horses. The maximum period of daily treatment that had been evaluated previously was 14 days. Hence, to be sure we treated long enough to detect a potential effect, we doubled the duration of daily treatments to 40 with visual monitoring and blood collections encompassing before, during, and after treatment. We wanted to mimic as close as possible the routine and clinical application to determine when if any side effects could be detected during or after treatment to have some estimate of number of safe treatment days for future studies.
We decided to use oral medication instead of intravenously because at the time the experiment was developed there was not a commercial intravenous firocoxib product in Brazil; furthermore, we believe that oral administration is less invasive, safer and more practical way to be administered by anyone. We ensured that the whole dose was consumed by each animal based on monitoring by the research team who confirmed the entirety of food and treatment dose were consumed daily. We understand the concern about the absorption but, again, this study was not designed as a full exposure study. That aspect will be addressed in future studies to fully evaluate in fact the absorption was not measured, but the animal intake was pretty much observed and guaranteed.
- Blood results and tables.
Routinely ALT is not measured in the horse and is not a reliable indicator of hepatic injury. It is not a viable enzymatic marker in the horse, please remove. Did you consider looking at liver functionality such as serum bile acids. This would add more validity to your manuscript as far as assessing functionality of liver mass in each horse. Also were these samples run only once or were they run in duplicate or triplicate to assess for variation. If they were not please state in the manuscript.
- Agreed, ALT was removed.
We agree that looking for liver functionality such as bile acids would add more information; however, assessment of these endpoints beyond the scope of this preliminary or pilot study but will be important in a more comprehensive future study.
Added to text “Various hematological and biochemical endpoints were determined by single-sample analysis.”.
The tables are very hard to understand from a statistical significance because it does not match up to the reference range provided for each analyte. This was for erythrocytes, hemoglobin, PCV, leukocytes, TP, AST, ALP and creatinine. All of these while stated were outside of the SEM but were still within the stated reference range so how significant is this really. This needs to be addressed better in the discussion. Also, you have to consider for liver enzymatic activity when it decreases this clinically means nothing, this is also the same consideration for creatinine. Please revise this portion of the discussion.
- Discussion have been revised to address the points suggested by the reviewer:
“In summary, the long firocoxib treatment caused changes in erythrocytes, hemoglobin, PCV, leukocytes, TP, AST, ALP and creatinine when compared to basal levels, but were still within the stated reference range so there is not a clinical significant change in fact. Also, the decreased liver enzymatic activity (AST and ALP) and creatinine does not show meaningful considerations and is only stated to be faithful to the findings. So, daily oral administration of firocoxib (0.11-0.14 mg/kg) as a selective NSAID COX-2 inhibitor for 40 days to healthy adult male and female horses resulted in some hematological or biochemical effects during treatment that returned to pre-treatment or baseline values post-treatment without any clinically relevant adversity. While firocoxib appeared favorable for prolonged use in horses, future studies are required to reproduce these results with other more comprehensive study designs that may involve prolonged administration to heathy as well as injured or diseased animals to evaluate efficacy and use of other clinically relevant endpoints.”
Globally this manuscript just provides descriptive information about longterm use of firocoxib and potential hematological effects on a very small sample size. Please consider adding more animals or justification for the size of the sample pool. While there may be some clinical value in knowing that we may be able to leave horses on this medication for an extended period of time with minimal side effects in the current state of this manuscript and how it is presented that cannot be adequately determined without additional information and revision.
- The authors agree about the importance of testing the medication in a larger number of animals, especially those that may be injured or diseased and with the addition of other relevant variables, However, as a preliminary or pilot study, the objective and hypothesis of this study has been stated using a statistically sufficient number of healthy adult horses. While the results of this study were favorable, they provide a basis to design a more comprehensive study involving the prolonged use of the medication to address its efficacy. In the meantime, the practical application of the approach and favorable outcomes, support the prolonged use of firocoxib and anti-inflammatory treatment in horses.

Reviewer 4 Report
Comments and Suggestions for Authors
Dear authors,
You are dealing with an interesting subject, especially from a clinical point of view.
Although your results are clearly presented, I feel that the methodology could be improved. I was surprised to see that there were no controls in your study. That definitely affects the soundness of your results. Moreover, I would expect to see more variables e.g. an effort to estimate blood loss in the feces, or SDMA or bile acid concentrations. Finally, concerning the intake of the drug, administration through a nasogastric tube would have been a safer method.
Comments on the Quality of English LanguageThe quality of English language is good. I could spot only one spelling mistake:
Line 42: osteoarthristis should be corrected to osteoarthritis
Author Response
The authors appreciate the comments and grateful for the recognition of the importance of subject. Some clarifications are needed. We used the basal values of each healthy horse as their own control. Hence, sample collections before treatment (D0) was used as the point to begin treatment and basis for comparison to evaluate the effects of prolonged treatment.
The authors also understand the value of evaluation of blood loss in feces and that SDMA or bile acid concentrations would have given additional information. However, these variable were not included in the initial design of the experiment but will be included in a more comprehensive follow up experiment of ours.
The administration firocoxib through a nasogastric tube would result in direct exposure in the stomach, which was not practical. Moreover, administration in this manner for 40 consecutive days would result in excessive manipulation to the nostrils, esophagus, and stomach and likely cause undue stress to the animals. For these reasons, we believe that oral administration under supervision of intake was the less stressful way to give the firocoxib orally in a more practical manner to the horses.
Reviewer 5 Report
Comments and Suggestions for Authors
Manuscript is well written and provides clinically usefully information to equine practitioners.
Author Response
- Thank you very much for your considerations and we really appreciate your final comments about our work.